# Hiltonol Cocktail Kills Lung Cancer Cells by Activating Cancer-Suppressors, PKR/OAS, and Restraining the Tumor Microenvironment

**DOI:** 10.3390/ijms22041626

**Published:** 2021-02-05

**Authors:** Shu-Chun Chang, Bo-Xiang Zhang, Emily Chia-Yu Su, Wei-Ciao Wu, Tsung-Han Hsieh, Andres M. Salazar, Yen-Kuang Lin, Jeak Ling Ding

**Affiliations:** 1The PhD Program for Translational Medicine, College for Medical Science and Technology, Taipei Medical University, 250 Wusing Street, Taipei 110, Taiwan; b101106043@tmu.edu.tw; 2International Ph.D. Program for Translational Science, College of Medical Science and Technology, Taipei Medical University, 250 Wusing Street, Taipei 110, Taiwan; 3Graduate Institute of Biomedical Informatics, College of Medical Science and Technology, Taipei Medical University Hospital, 252 Wusing Street, Taipei 110, Taiwan; emilysu@tmu.edu.tw; 4Clinical Big Data Research Center, Taipei Medical University Hospital, 252 Wusing Street, Taipei 110, Taiwan; 5Graduate Institute of Medical Sciences, College of Medicine, Taipei Medical University, 250 Wusing Street, Taipei 110, Taiwan; d119107002@tmu.edu.tw; 6Department of Thoracic Surgery, Department of Surgery, Taipei Medical University Shuang Ho Hospital, Taipei 110, Taiwan; 7Joint Biobank, Office of Human Research, Taipei Medical University, 250 Wusing Street, Taipei 110, Taiwan; thhsieh@tmu.edu.tw; 8Oncovir, Inc., 3203 Cleveland Avenue Northwest, Washington, DC 20008, USA; asalazar@oncovir.com; 9Big Data Research Center, Taipei Medical University, 250 Wusing Street, Taipei 110, Taiwan; robbinlin@tmu.edu.tw; 10Biostatistics Center, Office of Data Science, Taipei Medical University, 250 Wusing Street, Taipei 110, Taiwan; 11Graduate Institute of Data Science, College of Management, Taipei Medical University, 250 Wusing Street, Taipei 110, Taiwan; 12Department of Biological Sciences, National University of Singapore, Singapore 117543, Singapore

**Keywords:** NSCLC (non-small cell lung cancer), combinatorial treatment with Hiltonol^+++^ cocktail [Hiltonol+anti-IL6+stattic+AG490], tumorigenic microenvironment, anti- and pro-tumorigenic cytokine production, tumor-suppressors PKR (protein kinase R) and OAS (2′5′ oligoadenylate synthetase)

## Abstract

NSCLC (non-small cell lung cancer) is a leading cause of cancer-related deaths worldwide. Clinical trials showed that Hiltonol, a stable dsRNA representing an advanced form of polyI:C (polyinosinic-polycytidilic acid), is an adjuvant cancer-immunomodulator. However, its mechanisms of action and effect on lung cancer have not been explored pre-clinically. Here, we examined, for the first time, how a novel Hiltonol cocktail kills NSCLC cells. By retrospective analysis of NSCLC patient tissues obtained from the tumor biobank; pre-clinical studies with Hiltonol alone or Hiltonol^+++^ cocktail [Hiltonol+anti-IL6+AG490 (JAK2 inhibitor)+Stattic (STAT3 inhibitor)]; cytokine analysis; gene knockdown and gain/loss-of-function studies, we uncovered the mechanisms of action of Hiltonol^+++^. We demonstrated that Hiltonol^+++^ kills the cancer cells and suppresses the metastatic potential of NSCLC through: (i) upregulation of pro-apoptotic Caspase-9 and Caspase-3, (ii) induction of cytosolic cytochrome *c*, (iii) modulation of pro-inflammatory cytokines (GRO, MCP-1, IL-8, and IL-6) and anticancer IL-24 in NSCLC subtypes, and (iv) upregulation of tumor suppressors, PKR (protein kinase R) and OAS (2′5′ oligoadenylate synthetase). In silico analysis showed that Lys296 of PKR and Lys66 of OAS interact with Hiltonol. These Lys residues are purportedly involved in the catalytic/signaling activity of the tumor suppressors. Furthermore, knockdown of PKR/OAS abrogated the anticancer action of Hiltonol, provoking survival of cancer cells. Ex vivo analysis of NSCLC patient tissues corroborated that loss of PKR and OAS is associated with cancer advancement. Altogether, our findings unraveled the significance of studying tumor biobank tissues, which suggests PKR and OAS as precision oncological suppressor candidates to be targeted by this novel Hiltonol^+++^ cocktail which represents a prospective drug for development into a potent and tailored therapy for NSCLC subtypes.

## 1. Introduction

Lung cancer is a leading cause of cancer deaths worldwide [1], with metastasis being the main driving force of fatality [2]. Since lung cancer is often diagnosed late, treatment options such as surgery, radiation therapy, and targeted chemotherapy may provide only limited outcomes. Despite improvements in diagnosis and therapy over several decades, the prognosis of lung cancer patients remains dismal with 5-year survival rate of only ~20%. NSCLC (non-small cell lung cancer), which accounts for about 85% of all lung cancers, is relatively insensitive to chemotherapy. Therefore, there is urgency to identify specific regulators and to understand how they might mediate anticancer effects and cell death in NSCLC cells, with a view to developing precision diagnostics and targeted cancer therapeutics.

PolyI:C (polyinosinic-polycytidilic acid) has been shown to suppress the proliferation and survival of lung cancer cells [3]. Hiltonol (also known as poly-ICLC) is an advanced polyI:C synthetic RNA designed by stabilizing polyI:C with poly-lysine. Hiltonol has been shown to induce immune response against established tumors [4,5]. However, understanding the mechanism of anti-tumorigenic action of Hiltonol, especially that of lung cancer is important, albeit hitherto unexplored. Thus far, Hiltonol has only been applied as an investigational adjuvant to stimulate immune cells such as CD8-T cells and dendritic cells in cancer patients. This is despite recent clinical trial on lung cancer patients (https://immuno-oncologynews.com/2018/05/29/first-lung-cancer-patient-treated-with-neo-pv-01-phase-1-trial/ (accessed on 6 October 2020)), which was designed to evaluate the safety and tolerability of NEO-PV-01, administered alongside Hiltonol, Keytruda, and Alimta (pemetrexed) and Paraplatin (carboplatin).

Lung cancer cells have been shown to be susceptible to polyI:C combined with inhibitors of IL6 and JAK2/STAT3 [3]. Here, we aimed to investigate whether and how Hiltonol, an advanced form of polyI:C, plays an antitumorigenic role in human lung cancer. Gaining insights on the mechanisms underlying the advantage of Hiltonol and Hiltonol cocktail [Hiltonol+anti-IL6+AG490 (JAK2 inhibitor)+Stattic (STAT3 inhibitor)] is essential for the development of anticancer drug cocktail with improved specificity and avoidance of off-target effects, since NSCLC is notorious for its heterogeneity.

We discovered that a low dose of Hiltonol potently suppressed and killed the cancer cells. Compared to polyI:C, we found that Hiltonol exerted greater apoptosis. Of significance, Hiltonol activated two dsRNA-binding cancer-suppressor proteins, PKR (protein kinase R) and OAS (2′5′ oligoadenylate synthetase). Concordantly, knockdown of PKR (or OAS) abrogated Hiltonol-mediated killing and re-established NSCLC cell survival and proliferation. In silico analyses predicted interactions between Hiltonol and PKR (or OAS) indicating probable binding sites, involving Lys296 of PKR and Lys66 of OAS, which were earlier reported to be directly involved in the catalytic/signaling activity of these tumor suppressors [6,7,8]. Retrospective analysis of primary tissues from NSCLC patients showed marked reduction in PKR and OAS as the disease advanced, corroborating the pivotal roles of PKR and OAS in suppressing lung cancer. Notably, our results suggest that Hiltonol^+++^ not only exerted death on the NSCLC cells, it dramatically reduced the metastatic potential by mitigating their migration and invasion capacities. Hiltonol repressed pro-inflammatory cytokines and increased anticancer cytokine, IL-24 to different extents in different NSCLC subtypes. Neutralizing anti-IL-24 antibody inhibited the anti-cancer activity of IL-24, which was earlier induced by Hiltonol treatment. Taken together, we propose that Hiltonol^+++^ is a novel cocktail drug candidate, worthy of development for lung cancer treatment.

## 2. Results

### 2.1. Hiltonol^+++^ (Hiltonol+Anti-IL6+JAK2+STAT3 Inhibitors) Cocktail Effectively Killed Lung Cancer Cells

First, we investigated the effective dosage of Hiltonol from 10, 20, 50, to 100 μg/mL on four human NSCLC cell lines: A549, H292, H1299, and H358. We found that lower doses of 10–20 μg/mL exerted more efficacious activity in reducing cell viability (Figure 1A), in particular, against the highly invasive cell lines, H358 and H292. Concordantly, cancer treatment trials have shown that although high doses of Hiltonol upregulated the production of interferon and other cytokines, only modest therapeutic efficacy with moderate and transient cytotoxicity was achieved. Another pilot clinical trial showed that patients treated with a low dose of Hiltonol (10–50 μg/kg) displayed little or no cytotoxicity, but a preserved quality of life [9]. We found that up to 55% of the cancer cells reduced its viability within 72 h of treatment with 20 μg/mL Hiltonol (red line in Figure 1A). By contrast, treatment with polyI:C at the optimized concentration of 10 μg/mL (blue line) reduced viable cells up to only 35% (*p* < 0.01). We observed that Hiltonol^+++^ (Hiltonol in combination with inhibitors of IL6, JAK2, STAT3) significantly suppressed the viability of lung cancer cells compared to Hiltonol alone (Figure 1B). Hiltonol is the structurally improved and more stable form of polyI:C [10], and our results showed that Hiltonol exerts an enhanced anticancer effect compared to polyI:C. Furthermore, treatment with 20 μg/mL Hiltonol^+++^ for 72 h exerted greater killing of up to 85% of H358 and A549 cells, relative to a comparable dose of polyI:C^+++^ (*p* < 0.001) (Figure 1B).

### 2.2. Hiltonol^+++^ Suppressed Lung Cancer Cell Proliferation through Intrinsic Apoptosis

The imbalance between cell proliferation and cell death is a critical hallmark of tumorigenesis. Thus, we compared the impact between Hiltonol and polyI:C on cell proliferation. Both Alamar Blue and Trypan blue exclusion assays showed that 20 μg/mL Hiltonol alone was sufficient to attenuate cell proliferation of all four lung cancer cell lines (Figure 2A). Moreover, treatment with a single dose of 20 μg/mL Hiltonol^+++^ cocktail for 72 h inhibited cell proliferation by a further 32%, suggesting the involvement of IL6/JAK2/STAT3 signaling in lung cancer progression, since AG490 and stattic (present in Hiltonol^+++^ cocktail) inhibited Jak2 and Stat3, respectively. Hiltonol^+++^ cocktail also exerted a further reduction in cell growth by up to 50% compared to polyI:C^+++^ (an average taken from Alamar blue test and Trypan blue exclusion assay ***, *p* < 0.001). Notably, the slow growing but highly-invasive H358 cells were more sensitive to Hiltonol alone, suggesting that on its own, Hiltonol may preferentially suppress cell proliferation and induce apoptosis of H358 compared to other lung cancer cells. Next, double-staining with Annexin V and 7-AAD (7-aminoactinomycin) followed by FACS analysis revealed early apoptosis (Figure 2B) when treated with Hiltonol alone, compared to polyI:C alone. Treatment with Hiltonol^+++^ significantly elevated cell apoptosis (**, *p* < 0.01; ***, *p* < 0.001). Consistent with cell viability study (Figure 1B), A549 and H358 seemed more responsive to Hiltonol^+++^, displaying 87% and 77% early apoptosis, respectively. Thus, Hiltonol^+++^ is an efficacious drug cocktail candidate. Representative results of apoptosis assays are shown in Appendix A.

We found that treatments with Hiltonol significantly activated Caspases-9 and -3 (Figure 2C,D; **, *p* < 0.01; ***, *p* < 0.001). To confirm the contribution of Hiltonol on intrinsic apoptosis (and using polyI:C as a reference for comparison), we tested the release of cytochrome *c* in the treated NSCLC cells. Consistently, we found that Hiltonol alone increased the levels of cytosolic cytochrome *c* (Figure 2E, blue boxes). Hiltonol^+++^ further increased the cytosolic cytochrome *c* at the expense of mitochondrial cytochrome *c* (Figure 2E, red boxes). Altogether, these results affirmed that Hiltonol induced NSCLC cell apoptosis via the intrinsic mitochondria-associated pathway, especially against A549 and H358 variants, which are the most frequent AD (adenocarcinomas) subtypes of NSCLCs. Future studies may be performed with an in vivo senescence assay to exclude the possibility of Hiltonol-induced senescence instead of cell death.

### 2.3. Hiltonol^+++^ Suppressed the Metastatic Potential of Lung Cancer Cells

Since metastasis is the leading cause of cancer-related deaths, we evaluated the impact of Hiltonol^+++^ on cell migration and invasion. Figure 3A–D shows that Hiltonol^+++^ slowed down cell migration rate by 80% (***, *p* < 0.001), equivalent to 4-fold greater efficacy compared to polyI:C^+++^. Consistently, Hiltonol^+++^ significantly decreased the cancer cell invasion by up to 76% while polyI:C^+++^ only repressed cell invasion by up to 40% for different cell types (Figure 3E, Appendix A). Altogether, Hiltonol^+++^ effectively downregulated lung cancer metastasis potential.

### 2.4. Hiltonol Suppressed Pro-Inflammatory Cytokines but Upregulated Anticancer IL-24 in Lung Cancer Cells

Since Hiltonol was originally designed as an adjuvant immune stimulator, we investigated its impact on cytokine production (Figure 4A–G). We found that a low dose of Hiltonol significantly reduced pro-inflammatory cytokines: GRO, MCP-1, IL-8, and IL-6 (Figure 4A–D, red boxes) in H1299 and H358 cells, but these cytokines were increased (blue boxes) in A549 and H292 cells. These observations could be explained by tumor heterogeneity. On the other hand, polyI:C indiscriminately stimulated these cytokines in all four cancer cell lines.

The pro-inflammatory cytokines, GRO, MCP-1, IL-8, and IL-6, purportedly promote lung cancer development and progression [11,12,13]. It is conceivable that Hiltonol directly suppressed survival of H1299 and H358 cells by subduing these pro-inflammatory cytokines, hence restraining the tumorigenic microenvironment. Concordantly, we showed that treatment with recombinant proteins (500 pg/mL MCP-1 in H1299 and 1000 pg/mL IL-6 in H358) significantly restored the anti-cancer effects caused by Hiltonol (Figure 4H). Conversely, we observed that Hiltonol was more effective in elevating tumor suppressor, IL-24, in A549 and H292 cells (Figure 4E, green boxes). Furthermore, neutralizing anti–IL-24 polyclonal antibody inhibited the anti-cancer effects of IL-24 which was earlier induced by Hiltonol treatment (Figure 4I). As IL-24 is associated with downregulation of lung cancer metastasis [14], the Hiltonol-mediated increase in IL-24 supports the clinical potential of Hiltonol. IL-24 has been reported to mediate cytokine activities either dependently or independently of the JAK/STAT signaling pathway [15]. Here, we showed that Hiltonol^+++^ (containing AG490 and stattic which inhibits JAK/STAT), suppressed lung cancer cell growth and survival. However, the inhibition of JAK/STAT was accompanied by a marked increase in anticancer cytokine, IL-24. Therefore, it is conceivable that the anticancer activity of IL-24 suppressed lung cancer cells independently of JAK/STAT. Altogether, our data suggest that Hiltonol mediates an intricate immune network, which is cell-type specific with differential outcomes on the heterogeneous NSCLC cell types. The balance of immunomodulatory activities between pro- and anti-inflammatory cytokines probably underlies one of the multiple modes of Hiltonol-mediated killing of NSCLC cells.

### 2.5. Knockdown of PKR and/or OAS Abrogated Hiltonol-Mediated Killing of NSCLC Cells

To understand how Hiltonol kills lung cancer cells, we first examined MDA5 and TLR3, which are PRRs (pattern recognition receptors) that bind dsRNAs. This is of interest since Hiltonol (and polyI:C) are both dsRNAs which mimic viral dsRNAs. We found that polyI:C significantly upregulated MDA5 level in H292 and H1299 (Figure 5A) and boosted TLR3 expression in A549 cells (Figure 5B). On the other hand, Hiltonol mainly triggered MDA5 and TLR3 in H1299 cells, a lung cancer cell line which is apparently immutable and resistant to polyI:C (Figure 3A–D). This again indicates: (i) the superior anticancer potency of Hiltonol and (ii) that besides PRRs, other dsRNA-binding proteins also responded to Hiltonol treatment.

Hiltonol is reported to be the most potent stimulator of PKR (protein kinase R) and OAS (2′5′ oligoadenylate synthetase) in solid cancers [16]. PKR and OAS are enzymes which bind dsRNA, to suppress tumorigenesis. However, the association of PKR and OAS to Hiltonol treatment of lung cancer is hitherto unexplored, albeit pertinent to query. We found that Hiltonol effectively activated the mRNA expression of PKR in H292, H1299, and H358 cells (Figure 5C), and further boosted OAS mRNA expression in H292 and H358 cells (Figure 5D). Consistently, polyI:C upregulated PKR mRNA levels in A549, H1299, and H358 cells (Figure 5E), further corroborating the cancer-suppression role of PKR. Additionally, PKR is known to trigger apoptosis through FADD–mediated activation of caspase-8 [17,18], whereas OAS is required for IFNγ-induced apoptosis [19]. This prompted us to further investigate the actions of PKR and OAS in NSCLC during Hiltonol treatment. Knockdown of PKR (or OAS) using gene-specific RNAi was conducted in H292 and H358 cells, achieving up to 85% and 75% loss of mRNA levels in PKR and OAS, respectively (Appendix A; ***, *p* < 0.001). Figure 5F,G shows that knockdown of PKR (or OAS) reversed the Hiltonol-mediated reduction of cell viability in NSCLC (blue boxes), resulting in the recovery of cancer cell viability and proliferation in H292 and H358 cells. Double knockdown of PKR and OAS further nullified the anticancer effect of Hiltonol as the cancer cell viability and proliferation were re-established (red boxes). Control siRNA (and without Hiltonol treatment) displayed no impact on cell viability or proliferation (Appendix A). Therefore, knockdown of these two dsRNA-binding proteins seemed to abrogate the cancer cell-killing potency of Hiltonol, demonstrating that Hiltonol targets and apparently unleashes these two tumor-suppressors against lung cancer. Our studies provide explanations on how Hiltonol kills NSCLC through PKR and/or OAS.

### 2.6. In Silico Analysis Showed Interaction between Hiltonol and PKR/OAS

To analyze the potential molecular interaction between Hiltonol and PKR/OAS, we performed in silico predictions by AutoDock [20] and showed the potential amino acid residues (red) involved (Figure 6A–F). The binding structure of PKR (or OAS) to Hiltonol is shown in Figure 6B,E, respectively. Figure 6C shows the superimpositions of PKR (PDB: 6D3K) interaction with the core structure of Hiltonol (green). There are 14 residues in PKR, which were predicted to interact with Hiltonol: Ser275, Phe278, Gly279, Gln280, Val281, Val294, Lys296, Glu367, Cys369, Thr373, Ser418, Asn419, Phe421, and Asp432. Figure 6F shows the superimpositions of OAS (PDB: 4IG8) interaction with Hiltonol, predicting the following residues binding to Hiltonol: Ser63, Lys66, Thr68, Leu70, Arg73, Ser74, Asp75, Gln194, Lys213, Gln217, Pro228, Gln229, Tyr230, Glu233, and Leu308.

Interestingly, the two lysine residues in both the PKR and OAS, which were predicted to interact with Hiltonol, also harbor other important functions: (i) Lys296 in PKR (see underlined, Figure 6A) was reportedly important for PKR catalytic activity [17,18]; (ii) Lys66 in OAS (see underlined, Figure 6D) is known to be pivotal to the signaling activity of OAS-like protein [6]. These findings indicate that Lys296 of PKR and Lys66 of OAS may be directly interacting with Hiltonol to exert crucial structure–activity relationships. Future studies may include mutagenesis of these lysine residues in PKR and OAS to confirm their roles in Hiltonol-mediated activation of the tumor suppressors.

### 2.7. Tumor-Suppressors, PKR, OAS, and IL-24 Were Repressed in Advanced Stages of Primary Lung Cancer

To affirm the physiological significance of PKR, OAS, and IL-24 in lung cancer, we retrospectively examined human primary lung tissues by both TCGA database analyses and IHC (immunohistochemistry) staining. In total, *n* = 1116 primary samples were used for TCGA analysis, including AD (adenocarcinoma) tissues (*n* = 509), SCC (squamous cell carcinoma) tissues (*n* = 498) and normal tissues (*n* = 109). Fourteen pairs of lung carcinoma tissues and correspondingly paired control normal lung tissues were examined by IHC. Figure 7A shows that the PKR and OAS mRNA levels were significantly upregulated in AD tissues, although their protein levels were reduced in NSCLC tissues (Figure 7B). Appendix A summarizes the clinicopathological parameters of the lung cancer patients from whom lung tissue samples were obtained. The protein expression levels of PKR and OAS were substantially and progressively reduced in advanced stage carcinoma tissues (Figure 7C). A similar trend was observed with IL-24.

The data thus far indicate: (i) the tumor-suppressor functions of PKR and OAS, which correlate well with the anti-tumor properties of IL-24 in lung cancer and (ii) that expression of PKR and OAS might be regulated by post-transcriptional (or post-translational) modifications. This is further supported by computational analysis, which revealed that PKR contains extensive post-translational modification sites for phosphorylation, glycosylation, lipidation, and ubiquitination (Appendix A). For PKR, the predicted residues involved in N-/C-terminal lipidations are: Gly3, Gly8, Gly551 and the predicted residues involved in ubiquitination are: Lys134, Lys150, Lys173, Lys253, Lys261, and Lys352. OAS contains mainly phosphorylation sites. Similarly, IL-24 contains phosphorylation, glycosylation, and N-terminal lipidation (Cys18, Cys35) sites. Altogether, these computational analyses suggest that the residues of PKR, OAS, and IL-24 are post-translationally modified, and presumably contribute to the: (i) protein stability, (ii) auto-dimerization of PKR/OAS, and (iii) formation of the binding pocket for interactions with dsRNA molecules like Hiltonol.

Overall, investigation of the primary lung cancer tissues over advancing stages of the disease has confirmed the lung cancer cell line studies, supporting the roles of PKR, OAS, and IL-24 as specific lung cancer suppressor candidates, worthy of consideration for therapeutic strategies.

## 3. Discussion

Lung cancer is amongst the top cancer-related deaths worldwide, with an estimate of 1.6 million fatalities each year [24]. NSCLC accounts for 85% of all lung cancers, of which AD (adenocarcinoma) and SCC (squamous cell carcinoma) are the most frequent subtypes [25]. We previously reported that polyI:C^+++^ provoked apoptosis of lung cancer cells [3]. Here, we found that Hiltonol^+++^ effectively heightened the apoptosis of lung cancer cells by 4-fold, to reduce cell proliferation and survival. Hiltonol^+++^ appeared to exert stronger cell killing and inhibition of the metastatic potential of both the AD and SCC subtypes of NSCLC. We propose for the first time that Hiltonol^+++^ is a potent anticancer cocktail drug with potential for developing into an effective lung cancer therapy.

As a critical immune modulator, we found that a low dose of Hiltonol effectively killed lung cancer cells (Figure 1 and Figure 2), suppressed their metastasis (Figure 3), and upregulated the production of anticancer IL-24 in A549 and H292 cells, but downregulated pro-tumorigenic cytokines, GRO, MCP-1, IL-8, and IL-6 in H358 and H1299 cells (Figure 4). Thus, it is conceivable that Hiltonol restrained the tumorigenic microenvironment. Hiltonol appeared to stimulate different lung cancer subtypes to produce distinct cytokine profiles, plausibly due to the lung tumor cell heterogeneity. Heterogeneity is a main reason for drug-resistance during cancer treatment; therefore, an accurate assessment of the heterogeneous responses of lung cancer subtypes to a potential drug cocktail is essential for the development of effective and targeted therapies. Combinatorial approaches that target predominant and drug-sensitive cells seem likely to induce the most promising responses. Thus, elucidation of global cytokine profiles, in future, may provide further insights for fine-tuning ancillary developments of variant forms of Hiltonol-based combinatorial drugs against heterogeneous NSCLCs.

As a dsRNA, Hiltonol appears to display a significant advantage of target specificity, viz, while sparing other dsRNA-binding PRRs like MDA5 and TLR3. Hiltonol seemed to selectively activate PKR and OAS (two dsRNA-binding tumor-suppressor proteins) (Figure 5A–E), which killed the NSCLCs. Our retrospective study of lung cancer patient tissues corroborates the in vitro findings to enable molecular precision in defining PKR and OAS as critical tumor-suppressors of lung carcinogenesis (Figure 7B,C), and that their deficiency, as observed in the later stages of lung cancer patient tissues, likely provoked the advancement of the disease (Figure 7B,C). Functional domains analyzed by InterPro (https://www.ebi.ac.uk/interpro/ (accessed on 6 October 2020)) [22] indicated that PKR shows Ser/Thr kinase activity (amino acids 410–422), whereas OAS shows nucleotidyltransferase activity (amino acids 37–119) and 2′-5′-oligoadenylate synthetase activities (amino acids 62–81 and 295–305). Future studies may be directed to reveal how Hiltonol mediates the molecular structure-activities of PKR and OAS in lung cancer killing. These findings would further aid the refinement of Hiltonol for targeted therapy.

As binders of dsRNAs, PKR and OAS are very sensitive to the dose, length, and structure of dsRNAs. Since Hiltonol is a longer-chain stable dsRNA, it is anticipated to stimulate PKR and OAS more potently [26,27] than polyI:C. Thus, the anti-tumor activities of PKR and OAS, induced by Hiltonol (Figure 7D), explains why a low dose used in clinical trials was proficient and efficacious. The superiority of low-dose Hiltonol (over that of polyI:C) supports the importance of the optimum length and structure of the dsRNA, which specifically switched on PKR/OAS activities. Together with anti-IL6 and inhibitors of JAK2 and STAT3, the Hiltonol^+++^ cocktail simultaneously triggered anticancer cytokines to kill NSCLCs efficaciously (Figure 7E). On the other hand, PKR binds to dsRNA, resulting in a number of conformational changes, including potential for homodimerization. PKR homodimerization is known to lead to activation of PKR through rapid autophosphorylation of a stretch of amino acids, namely Ser242, Thr255, Thr258, Ser83, Thr88, Thr89, Thr90, Thr446, and Thr451 [21] (Figure 7D). Future studies may investigate which residues are involved in Hiltonol-associated binding activation to close in on the precise center of drug activity.

## 4. Conclusions

The specificity of Hiltonol and its efficacy at low dose in targeting and activating tumor-suppressors, PKR and OAS, plus the dampening of the tumor microenvironment offer a promising potential for developing Hiltonol^+++^ into a powerful and selective anticancer cocktail drug for lung cancer treatment. Such precision oncological determination is enabled by investigating tissues from the tumor biobank.

## 5. Materials and Methods

### 5.1. Tissue Samples

Fourteen cases of lung cancer patients and their paired normal lung tissues (in total, *n* = 28) were acquired from the BioBank, Taipei Medical University, Taiwan, after clinical diagnosis of lung cancer was confirmed by biopsy and histological evaluation. Primary tissues obtained include: 3-paired stage I patients, 4-paired stage II patients, 4-paired stage III patients, and 3-paired stage IV patients. Adjacent non-cancerous lung tissues were obtained at least 2 cm away from the tumor node. All lung cancer patient samples were collected from the BioBank, Taipei Medical University, Taiwan. Experiments were performed in accordance with institutional guidelines (Taipei Medical University–Joint Institutional Review Board; IRB: N201606009). The H & E staining was performed as previously described [28]. The primary sections were obtained from the BioBank, Taipei Medical University, Taiwan. Details for IHC staining are in the Appendix A. The stained tissue sections were observed under a microscope (Olympus, Tokyo, Japan) and images were acquired using software (EOS Utility, Canon, Uxbridge, UK).

### 5.2. Cell Lines and Reagents

Non-small cell lung cancer cell lines (A549, H292, H1299, and H358) were obtained from ATCC (American Type Culture Collection). A549 was cultured in complete Ham’s F-12K medium (Gibco, Waltham, MA, USA). H292 and H358 were cultured in complete RPMI 1640 medium (Gibco). H1299 was cultured in complete DMEM medium (Gibco). All the complete media were supplemented with 10% FBS (Thermo Scientific, Rockford, IL, USA) and 100 U/mL penicillin and 100 μg/mL streptomycin (Invitrogen). Hiltonol (poly-ICLC) was obtained from Oncovir, Inc., Washington, DC, USA. PolyI:C (polyinosinic-polycytidilic acid) is a synthetic short-chain analog of dsRNA from InvivoGen. Stattic (STAT3 inhibitor) and AG490 (JAK2 inhibitor) were from Sigma and Calbiochem, respectively. Rabbit anti-IL6 polyclonal antibody was ab6672, from Abcam, Cambridge, UK. Antibodies used in IHC or immunoblotting analysis were anti-PKR antibody (ab32052, Abcam), anti-OAS antibody (ab86343, Abcam), anti-IL24 antibody (ab115207, Abcam), GAPDH monoclonal antibody (Abcam), VDAC polyclonal antibody (Cell Signaling Technology, Inc., Danvers, MA, USA), and cytochrome *c* monoclonal antibody (BD Biosciences, San Jose, CA, USA). Recombinant proteins or neutralizing antibodies used in MTT or Trypan Blue exclusion analyses were recombinant human MCP-1 (279-MC, R&D systems), recombinant human IL-6 (206-GMP, R & D systems), and human IL-24 polyclonal antibody (AF1965, R & D systems).

### 5.3. Quantitative Real-Time PCR (qRT-PCR)

qRT-PCR was performed as previously described [28]. Details are in the Appendix A. Briefly, 24 h after treatment, total RNA was extracted with TRIzol reagent (Invitrogen), according to the manufacturer’s instructions. The primers were human TLR3 (135 bp product): sense, 5′-TCTCATGTCCAACTCAATCCA-3′; antisense, 5′-TGGAGATTTTCCAGCTGAACC-3′; human MDA5 (92 bp product): sense, 5′-CCGAGAGAAGATGATGTATAAAGCTA-3′, antisense, 5′-TTTGCATCTGTAATTCCAAAATCT-3′; human PKR (171 bp product): sense, 5′-ACTTTTTCCTGGCTCATCTC-3′, antisense, 5′-ACATGCCTGTAATCCAGCTA-3′; human OAS (150 bp product): sense, 5′-CAAGCTCAAGAGCCTCATCC-3′, antisense, 5′-TGGGCTGTGTTGAAATGTGT-3′; human GAPDH (131 bp product): sense, 5′-GTCTCCTCTGACTTCAACAGCG-3′, antisense, 5′-ACCACCCTGTTGCTGTAGCCAA-3′. All expression values were normalized based on GAPDH as an endogenous control.

### 5.4. Treatment of Lung Cancer Cells with Hiltonol and polyI:C, and Combinations with Anti-IL6, Stattic, and AG490

To investigate the therapeutic effect of Hiltonol, lung cancer cells were treated with Hiltonol at increasing concentrations of 10–100 μg/mL. PolyI:C was used at 10 μg/mL as previously described. The negative control was PBS (phosphate buffered saline). For combinatorial treatments, the complete medium contained 20 μM of AG490 (JAK2 inhibitor), 2 μM of Stattic (STAT3 inhibitor), and 200 μg/mL of anti-IL6 antibody, in DMSO. DMSO was the negative vehicle control. The cells under treatment were incubated for 24–48 h at 37 °C. 

### 5.5. Cell Viability Assay

Both MTT {3-(4,5-dimethylathiazol-2-yl)-2,5-diphenyl tetrazolium bromide} and CTB (CellTiter Blue) were used to determine the cell viability. MTT and CTB were purchased from Life Technologies (Carlsbad, CA, USA) and Promega (Madison, WA, USA), respectively. Experiments were performed as previously described [28]. To measure the viable cells due to functional mitochondrial activity at the end of Hiltonol or polyI:C alone, or Hiltonol^+++^ or polyI:C^+++^ (in combination with anti-IL6+stattic+AG490) treatments, the cells were incubated with either MTT reagent or CTB reagent. Details are in the Appendix A. Samples from each time point were normalized with corresponding PBS or DMSO controls. 

### 5.6. Cell Proliferation Assay

Both Alamar blue and Trypan Blue dye exclusion were used to measure the cell proliferation. The growth curves of cells were measured by Alamar blue (Invitrogen, Waltham, MA, USA), according to manufacturer’s instructions. The metabolically active cells were determined as described above. For Trypan Blue dye exclusion, the experiments were performed as previously described [28]. Details are in the Appendix A.

### 5.7. TCGA Database Analysis

TCGA database analysis was performed as previously described [28]. Details are in the Appendix A. Briefly, TCGA NSCLC transcript dataset was downloaded from R package, TCGAbiolinks. In total, 1116 primary tissues were analyzed, including 1007 carcinoma tissues (509 AD tissues and 498 SCC tissues) and 109 normal tissues. The indicated genes were analyzed using student’s *t*-test.

### 5.8. Computational Prediction of Hiltonol Binding Sites in PKR and OAS

PKR (accession number: P19525.2) and OAS (accession number: P00973) were analyzed for potential Hiltonol binding sites. AutoDock (http://autodock.scripps.edu/ (accessed on 6 October 2020)) was used for RNA-binding site prediction [20]. Hiltonol structurally contains 4 molecular component units (Polyinosinic acid, Glycolic acid, Cytosine arabinoside monophosphate and (4-Aminobutyl)carbamic acid). Polyinosinic acid showed the strongest binding efficiency with PKR (PDB: 6D3K), with only −7.8 Kcal/mol (free energy of binding). Cytosine arabinoside monophosphate showed the strongest binding efficiency with OAS (PDB: 4IG8), with only −7.2 Kcal/mol. Hence, Polyinosinic acid and Cytosine arabinoside monophosphate were selected for binding site prediction of PKR and OAS, respectively.

### 5.9. siRNA-Knockdown of PKR and OAS in NSCLC Cells

To measure the effects of PKR (and/or OAS) in Hiltonol–treated cells, H292 and H358 cells were transfected with siRNA for 24 h, followed by treatment with 20 μg/mL Hiltonol for 48 h. The cell viability and proliferation were analyzed. PKR siRNA (siGENOME Human EIF2AK2-SMARTpool) and OAS siRNA (siGENOME Human OAS1-SMARTpool) were purchased from Dharmacon (Lafayette, CO, USA). Control (scrambled) siRNA was purchased from Invitrogen. Both siGENOME Human EIF2AK2-SMARTpool and siGENOME Human OAS1-SMARTpool contained a mixture of 4 siRNAs provided as a single reagent, providing advantages in both potency and specificity, which was anticipated to achieve silencing effects of >75%. siRNA transfection into lung cancer cells (at 5 × 10^5^ cells per well of a 6-well plate) was conducted using Lipofectamine RNAiMAX (Invitrogen; Life Technology). The final concentrations of the siRNA mixtures contained one SMARTpool siRNA or two equimolar siRNAs in concentrations of up to 50 nM, according to the manufacturer’s instructions. Knockdown efficiency was examined using real time-PCR and immunoblotting (Appendix A) performed as previously described.

### 5.10. Cellular Apoptosis Assay

Apoptosis analysis was performed as previously described [28]. Details are in the Appendix A. Briefly, 12 h after the treatment, early apoptosis was measured using annexin V (BioLegend, San Diego, CA, USA) in conjunction with 7-AAD (BioLegend), according to manufacturer’s instructions. Early apoptosis (Annexin V^+^/7-AAD^−^) was then analyzed using FACScan flow cytometer.

### 5.11. Caspase-9 and -3 Assays

Caspase analysis was performed according to the manufacturer’s instructions. Details are in the Appendix A. Briefly, 24 h after the treatments of the lung cancer cells, apoptosis was examined by determining the caspase-specific cleavages of activated caspase-9 and -3 (Abcam). 

### 5.12. Cell Migration and Invasion Assays

Migration and invasion studies were performed as previously described [28]. Details are in the Appendix A. Briefly, 24 h after treatment of lung cancer cells with Hiltonol, polyI:C, PBS, Hiltonol^+++^, polyI:C^+++^, or DMSO control, cell migration assay was performed. For cell migration assay, 2-well Culture-Insert (ibidi, Martinsried, Germany) was applied according to the manufacturer’s instructions. For cell invasion assay, biocoat matrigel invasion chambers with 8-μm pores in 24-well plates (Corning, Discovery Labware, Inc., Bedford, MA, USA) were used according to the manufacturer’s instructions.

### 5.13. ELISA

To measure the effects of Hiltonol (or polyI:C) on cytokine release, the lung cancer cell lines were treated with 20 μg/mL Hiltonol, 10 μg/mL polyI:C, or PBS control for 48 h. The cultured medium was collected and applied to the following ELISA assay. The levels of MCP-1, IL-8, IL-6, IL-10, and TNF-α, secreted by the cells, were quantified by using OptEIA human ELISA Set (BD Biosciences) according to the manufacturer’s instructions. The levels of secreted GRO and IL-24 were quantified by using human ELISA Kit (Abcam) according to the manufacturer’s instructions.

### 5.14. Statistical Analysis

Data are expressed as means ± S.D. from three independent experiments, with three replicates per sample/condition tested. Differences between averages were analyzed by two-way ANOVA (two-way analysis of variance) with interaction. Doses and cell lines are considered as two factors in the two-way ANOVA. LSD (Least Significant Difference) was calculated for interaction. The significance level was set at 0.05 and the *p*-values are marked as ** for *p* < 0.01 and *** for *p* < 0.001. The acquired data from FACS were analyzed with BD FACSuite^TM^ software (BD Biosciences, CA, USA). The relative migration rate indicated as % gap closure was calculated using Image J analysis software. All target signals from IHC were quantified by HistoQuest software (TissueGnostics, Vienna, Austria). 

## Figures and Tables

**Figure 1 ijms-22-01626-f001:**
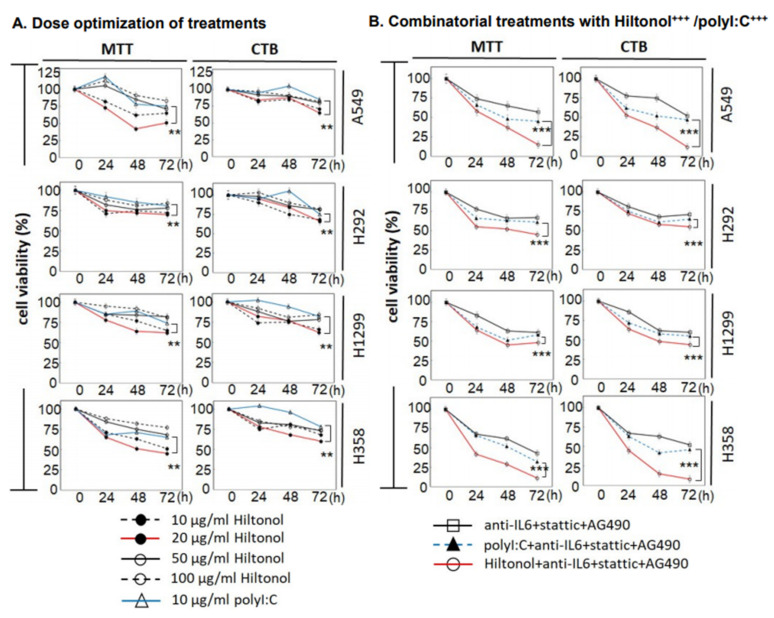
Treatment of human lung cancer cell lines (A549, H292, H1299, and H358) with Hiltonol+anti-IL6+stattic+AG490 (Hiltonol^+++^) significantly suppressed cancer cell survival. (**A**) Dose optimization of Hiltonol and polyI:C (polyinosinic-polycytidilic acid) was performed and cell survival was assayed using MTT {3-(4,5-dimethylathiazol-2-yl)-2,5-diphenyl tetrazolium bromide} and CTB (CellTiter Blue). Both assays consistently showed that Hiltonol significantly suppressed the cell viability. The optimal killing dose of Hiltonol is 20 μg/mL (red line), compared to polyI:C (blue line) (**, *p* < 0.01). Differences between averages were analyzed by two-way ANOVA (two-way analysis of variance) with interaction. Doses and cell lines were considered as two factors in the two-way ANOVA. In MTT assays, LSD (Least Significant Difference) value in A549, H292, H1299, and H358 are 3.902, 3.334, 2.172, and 2.046, respectively. In CTB assays, LSD value in A549, H292, H1299, and H358 are 3.063, 2.939, 2.28, and 1.739, respectively. (**B**) Combinatorial treatments with Hiltonol^+++^ (red line) or with polyI:C^+++^ (blue line) further reduced cancer cell survival, with up to 1.7-fold greater killing by Hiltonol^+++^ compared to polyI:C^+++^ (***, *p* < 0.001). Data are representative of means ± SD (*n* = 3). For each time point, cell counts were normalized to corresponding DMSO (or PBS (phosphate buffered saline))-treated controls. In MTT assays, the LSD value in A549, H292, H1299, and H358 are 3.745, 2.341, 3.004, and 2.489, respectively. In CTB assays, the LSD value in A549, H292, H1299, and H358 are 3.222, 2.969, 3.423, and 3.396, respectively.

**Figure 2 ijms-22-01626-f002:**
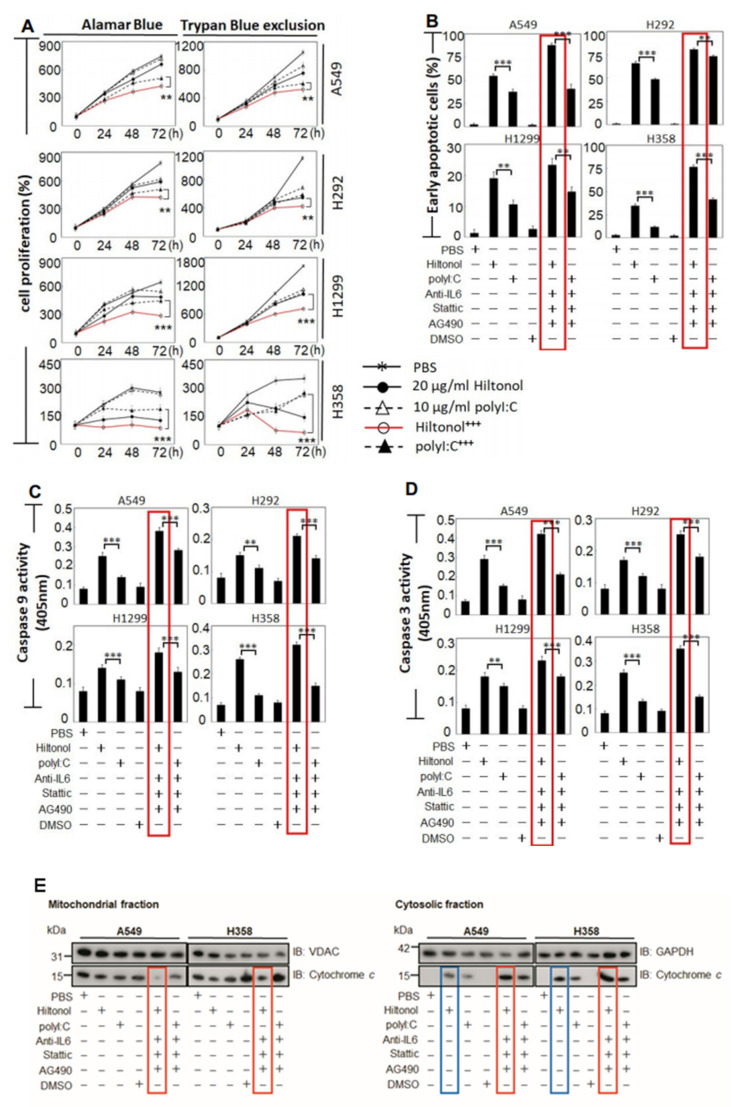
Treatments with Hiltonol^+++^ dramatically reduced NSCLC (non-small cell lung cancer) cell proliferation and increased early apoptosis of human lung cancer cells. (**A**) The dynamics of cell proliferation were measured using Alamar Blue or Trypan Blue dye exclusion tests. Both Hiltonol and polyI:C significantly decreased cancer cell proliferation. Hiltonol^+++^ (Hiltonol+anti-IL6+stattic+AG490) further significantly reduced cancer cell proliferation (red line), particularly for H358 cells. The effects of anti-IL6 or Stattic or AG490, or anti-IL6+Stattic+AG490 in NSCLC are shown in Figure 1B and previously in [3]. For each time point, cell counts were normalized with corresponding DMSO vehicle (or PBS) -treated controls. In Alamar Blue assays, the LSD value in A549, H292, H1299, and H358 are 20.486, 24.695, 28.913, and 20.445, respectively. In Trypan Blue exclusion studies, LSD value in A549, H292, H1299, and H358 are 42.633, 26.711, 22.67, and 19.413, respectively. **, *p* < 0.01; ***, *p* < 0.001. (**B**) Annexin V and 7-AAD (7-aminoactinomycin) double staining shows significant increase in early apoptosis (Annexin V^+^/7-AAD^−^) when treated with Hiltonol^+++^ (red boxes). LSD value in A549, H292, H1299, and H358 are 2.918, 2.009, 2.559, and 2.048, respectively. **, *p* < 0.01; ***, *p* < 0.001. Representative histograms of apoptosis are shown in Appendix A. (**C**,**D**) Caspases -9 and -3 activities in cells treated with Hiltonol alone or polyI:C alone or Hiltonol^+++^ or polyI:C^+++^ (polyI:C+anti-IL6+stattic+AG490) for 24 h showed that Hiltonol^+++^ caused the highest elevation in Caspases -9 and -3 activities (red boxes). Data are representative of means ± SD (*n* = 3). **, *p* < 0.01; ***, *p* < 0.001. In Caspase-9 studies, LSD value in A549, H292, H1299, and H358 are 0.002, 0.002, 0.002, and 0.002, respectively. In Caspase-3 studies, LSD value in A549, H292, H1299, and H358 are 0.001, 0.002, 0.001, and 0.001, respectively. (**E**) Cytochrome *c* release was examined to determine the levels of intrinsic apoptosis, which were increased in cells treated with Hiltonol alone or polyI:C alone or Hiltonol^+++^ or polyI:C^+++^.

**Figure 3 ijms-22-01626-f003:**
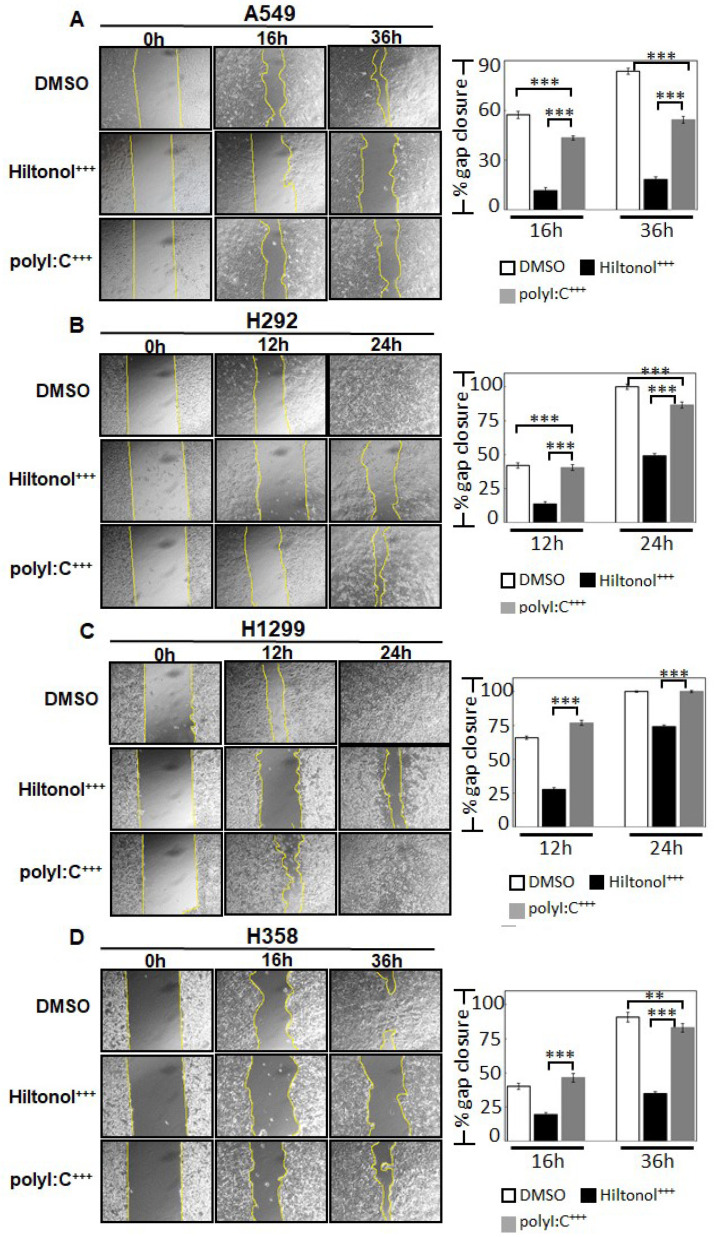
Hiltonol^+++^ efficiently suppressed lung cancer migration and invasion. The effects of Hiltonol alone or polyI:C alone or their respective combinatorial treatments (with anti-IL6+stattic+AG490) on metastatic potential in lung cancer were examined by migration assay: (**A**) A549 (**B**) H292 (**C**) H1299 and (**D**) H358 cells. The rate of migration was quantified as % gap closure normalized to corresponding 0 h time point DMSO-treated controls. Compared with polyI:C^+++^ (polyI:C+anti-IL6+stattic+AG490)-treated cells, the Hiltonol^+++^ (Hiltonol+anti-IL6+stattic+AG490)-treated cells displayed significant delay in migration, particularly for A549, H292, and H358 cells. In earlier time points (12 h or 16 h), the LSD value in A549, H292, H1299, and H358 are 1.876, 2.048, 1.697, and 2.247, respectively. In later time points (either 24 h or 36 h), the LSD value in A549, H292, H1299, and H358 are 2.17, 1.724, 0.83, and 2.799, respectively. **, *p* < 0.01; ***, *p* < 0.001. (**E**) shows cell invasion of these three cell lines. PolyI:C^+++^ reduced invasion of A549 and H358 by 30% and 12%, respectively, whereas Hiltonol^+++^ further abrogated cell invasion of A549, H292, and H358 cells by 57%, 36%, and 76%, respectively. ***, *p* < 0.001. The LSD value in A549, H292, and H358 are 10.187, 9.967, and 11.004, respectively.

**Figure 4 ijms-22-01626-f004:**
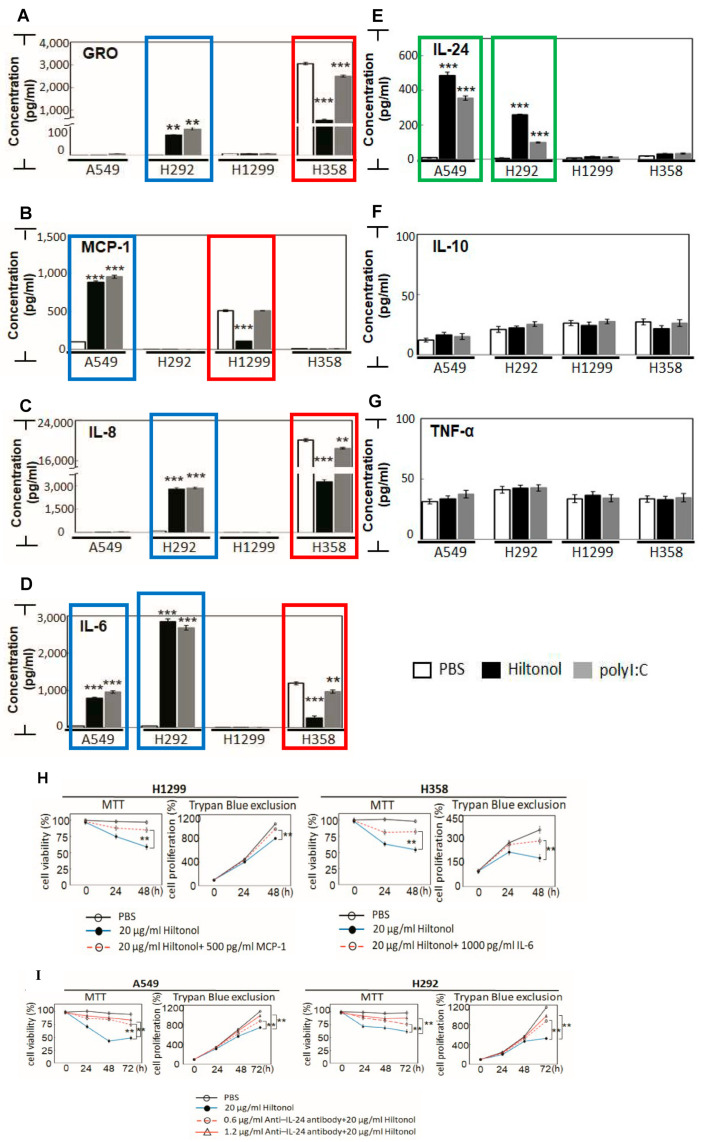
Hiltonol restrains tumorigenic microenvironment in human lung cancer cells. Key circulating pro- /anti-inflammatory cytokines were examined, including: (**A**) GRO, (**B**) MCP-1, (**C**) IL-8, (**D**) IL-6, (**E**) IL-24, (**F**) IL-10, and (**G**) TNF-α. Hiltonol treatment of A549 and H292 cells significantly increased the levels of GRO, MCP-1, IL-8, and IL-6 (blue boxes). The A549 and H292 normally express low levels of endogenous TLR3. Hiltonol treatment suppressed GRO, MCP-1, IL-8, and IL-6 (red boxes) in H1299 and H358 cells, which are known to normally express medium-to-high levels of TLR3 [3]. Compared with polyI:C treatment, Hiltonol is more effective in suppressing the pro-inflammation cytokines, e.g., GRO, IL-8, and IL-6 in H358 cells; as well as MCP-1 in H1299 cells. Consistently, Hiltonol-treated A549 and H292 cells displayed a significant activation of anticancer cytokine, IL-24 (green boxes). (**A**) The LSD value in A549, H292, H1299, and H358 are 1.436, 2.449, 0.566, and 35.328, respectively. (**B**) The LSD value in A549, H292, H1299, and H358 are 13.285, 0.436, 7.845, and 5.822, respectively. (**C**) The LSD value in A549, H292, H1299, and H358 are 31.385, 0.136, 5.805, and 67.348, respectively. (**D**) The LSD value in A549, H292, H1299, and H358 are 23.301, 40.908, 0.507, and 20.57, respectively. (**E**) The LSD value in A549, H292, H1299, and H358 are 10.761, 3.724, 5.284, and 17.336, respectively. (**F**) The LSD value in A549, H292, H1299, and H358 are 7.681, 4.1, 3.441, and 3.997, respectively. (**G**) The LSD value in A549, H292, H1299, and H358 are 3.883, 2.754, 1.884, and 2.023, respectively. (**H**) Recombinant proteins (MCP-1 in H1299 and IL-6 in H358) effectively restored the anti-cancer effects caused by Hiltonol. In MTT assays, the LSD value in H1299 and H358 are 3.633 and 3.537, respectively. In Trypan Blue exclusion studies, LSD value in H1299 and H358 are 17.225 and 20.688, respectively. (**I**) Neutralizing anti–IL-24 antibody inhibited IL-24 activity and significantly restored the cancer suppressive activity of Hiltonol in A549 and H292. In MTT assays, the LSD value in A549 and H292 are 3.89 and 3.443, respectively. In Trypan Blue exclusion studies, LSD value in A549 and H292 are 24.057 and 26.914, respectively. Data are representative of means ± SD (*n* = 3). **, *p* < 0.01; ***, *p* < 0.001.

**Figure 5 ijms-22-01626-f005:**
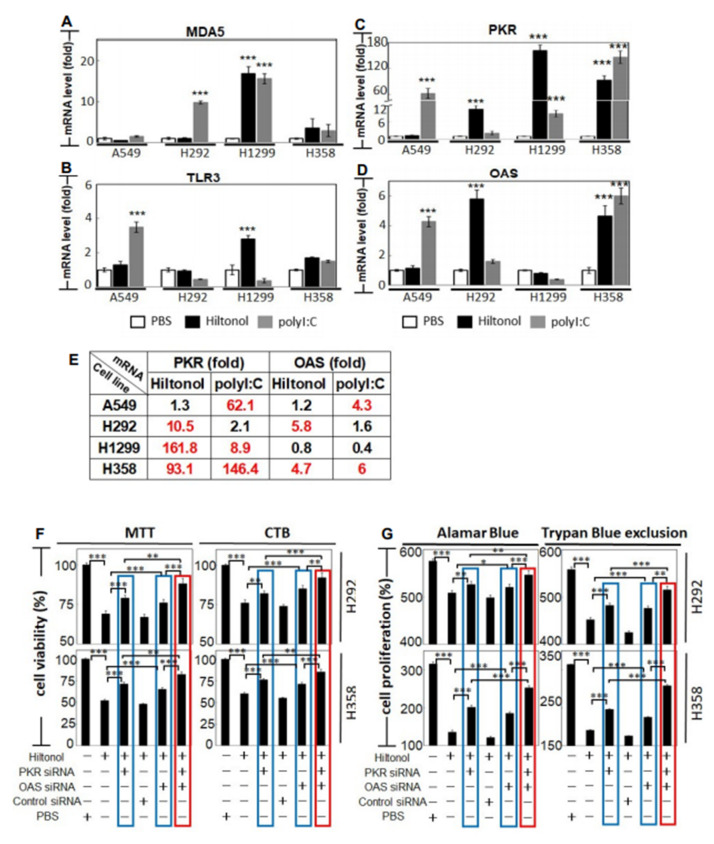
**Hiltonol significantly activated tumor suppressors, PKR and/or OAS, in human lung cancer cells.** The mRNA expression levels of: (**A**) MDA5 and (**B**) TLR3, (**C**) PKR (p68 protein kinase) and (**D**) OAS (2’5’ oligoadenylate synthetase) were measured. (**A**) LSD value in A549, H292, H1299, and H358 are 0.485, 0.227, 0.862, and 2.028, respectively. (**B**) LSD value in A549, H292, H1299, and H358 are 0.177, 0.319, 0.127, and 0.576, respectively. (**C**) LSD value in A549, H292, H1299, and H358 are 4.991, 0.679, 6.071, and 9.478, respectively. (**D**) LSD value in A549, H292, H1299, and H358 are 0.178, 0.528, 0.371, and 0.327, respectively. (**E**) shows the fold-change in the mRNA levels of PKR and OAS in response to Hiltonol (or polyI:C). The significant changes in the PKR and/or OAS mRNA overexpression in lung cancer cell lines are in red. To measure the effects of PKR and/or OAS in lung cancer upon Hiltonol treatment, 24 h after siRNA-knockdown of PKR or OAS or PKR+OAS double knockdown, NSCLC cells were treated with Hiltonol (or PBS) for 48 h, followed by viability and proliferation studies. (**F**) Cell survival was assayed using MTT and CTB. For each treatment, cell counts were normalized with PBS-treated controls. In MTT assays, LSD value in H292 and H358 are 2.87 and 2.631, respectively. In CTB assays, LSD value in H292 and H358 are 3.102 and 2.801, respectively. (**G**) Cell proliferation was measured using Alamar Blue or Trypan Blue dye exclusion tests. For each treatment, cell counts on 48 h post-treatment of Hiltonol (or PBS) were normalized with the corresponding 0 h treatment controls. PKR (or OAS) -specific siRNA significantly restored Hiltonol-induced cancer cell death and inhibited proliferation in H292 and H358 cells (blue boxes). PKR and OAS double knockdown further improved recovery of cancer cell viability and proliferation during Hiltonol treatment (red boxes). Data are representative of means ±SD (*n* = 3). *, *p* < 0.05; **, *p* < 0.01; ***, *p* < 0.001. In Alamar Blue assays, LSD value in H292 and H358 are 5.934 and 4.628, respectively. In Trypan Blue exclusion studies, LSD value in H292 and H358 are 4.978 and 1.743, respectively.

**Figure 6 ijms-22-01626-f006:**
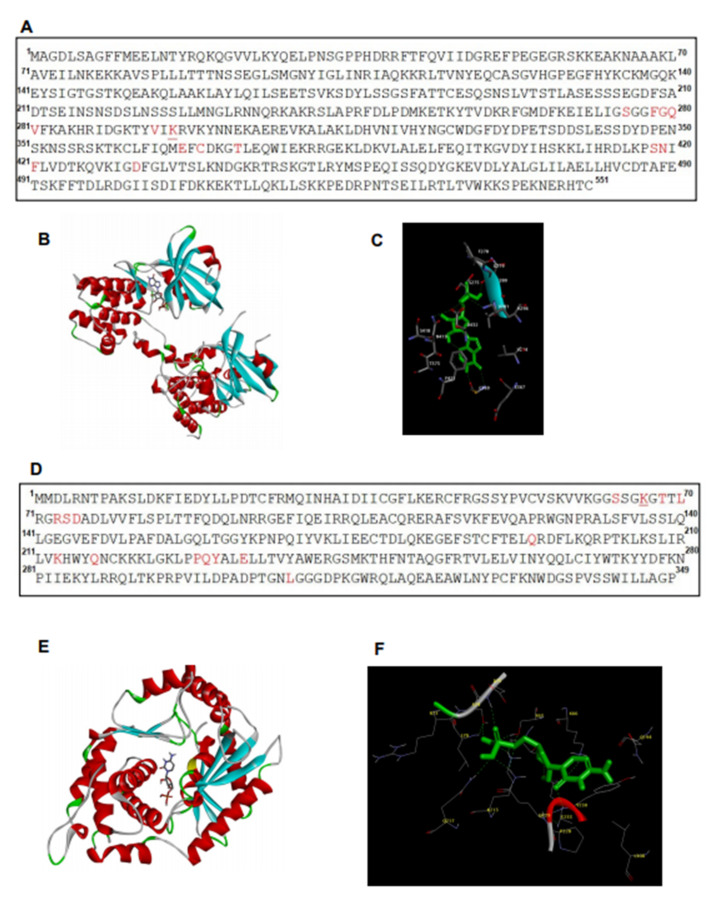
The predicted key residues of PKR and OAS involved in Hiltonol-binding. Potential Hiltonol-interacting residues and binding structures of: (**A**–**C**) PKR and (**D**–**F**) OAS are analyzed by AutoDock. The full structures of PKR (PDB: 6D3K) and OAS (PDB: 4IG8) were used for analysis. (**A**,**D**) The key residues of PKR and OAS involved in Hiltonol-binding are highlighted in red. The model shows that Lys296 of PKR and Lys66 of OAS (underlined) interact with Hiltonol, which is consistent with earlier reports [6,7,8]. The binding structures of PKR-Hiltonol and OAS-Hiltonol are shown in (**B**,**E**), respectively. The α-helices and β-sheets are colored red and blue, respectively. The coils are in “light grey” and the turns are in “green”. (**C**,**F**) show superimpositions of the binding poses of PKR and OAS with Hiltonol (green). Nitrogen and oxygen atoms are depicted in blue and red, respectively.

**Figure 7 ijms-22-01626-f007:**
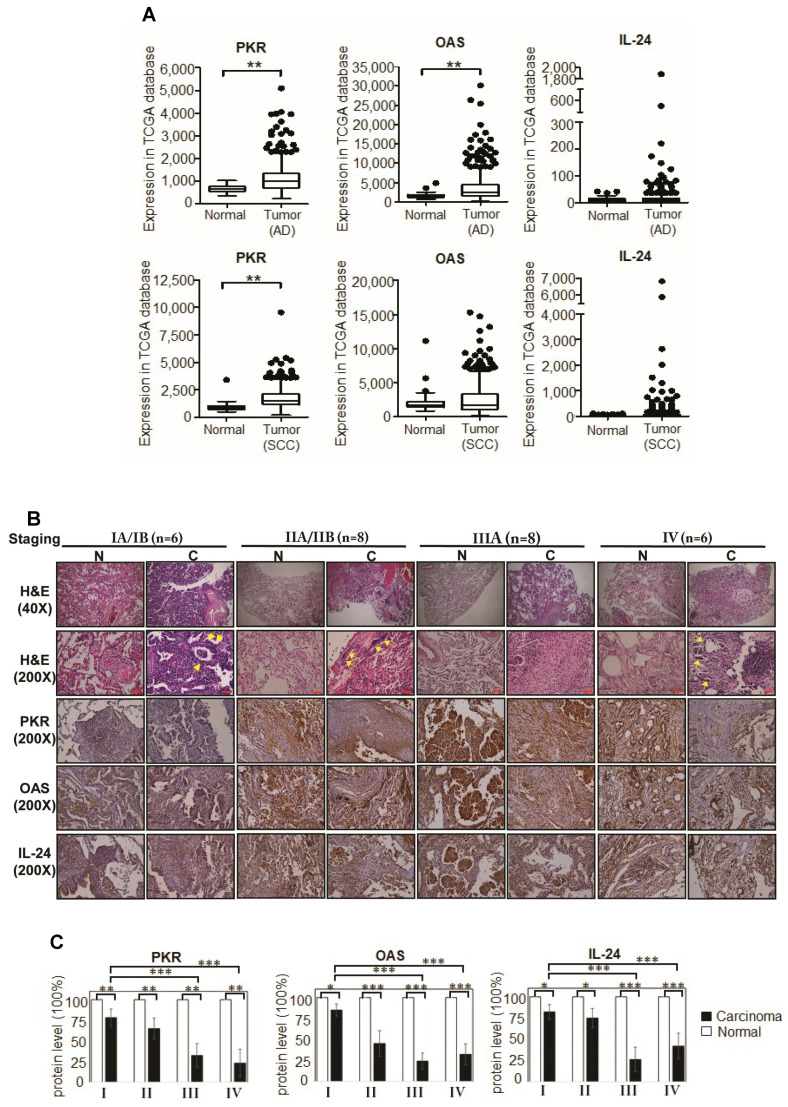
PKR, OAS, and IL-24 proteins which are expressed in normal tissues are suppressed in human primary lung cancer tissues. (**A**) Box-whisker plot showing the transcript levels of PKR, OAS, and IL-24 in TCGA database, respectively. Both AD (adenocarcinoma) tissues (*n* = 509), SCC (squamous cell carcinoma) tissues (*n* = 498) and normal tissues (*n* = 58 for AD group and *n* = 51 for SCC group) were used for analysis. Data are reported as normalized counts as provided in the TCGA level 3 data (**, *p* < 0.01). (**B**) Retrospective staining of patient lung cancer (“C”) and paired normal (“N”) tissues for PKR, OAS, and IL-24 was performed for lung cancer stages I, II, III, and IV. Clinicopathological parameters are shown in Appendix A. Inflammatory cell infiltration is indicated with yellow arrows. The protein levels of tumor suppressor markers, PKR and OAS, are dramatically reduced in carcinoma tissues, compared to normal controls. Consistently, anticancer cytokine, IL-24, showed a clear reduction in lung cancer tissues. Scale bar; 100 μm. (**C**) The corresponding quantitative results of IHC (immunohistochemistry) of PKR, OAS, and IL-24 are shown. (**D**) A hypothetical model to illustrate PKR activation via interaction with Hiltonol. There are 9 key residues involved in dsRNA-binding [21]. Computational analysis by InterPro (https://www.ebi.ac.uk/interpro/ (accessed on 6 October 2020)) [22] shows that N-terminal of PKR (brown) contains two dsRNA-binding motifs (a.a. 10-76 and a.a. 101-166), whereas C-terminal of PKR (atrovirens) is characterized mainly as protein kinase domain with Ser/Thr kinase active site (a.a. 410–422). The left panel shows that in the absence of Hiltonol, PKR remains in an inactive state, hence leading to tumor progression. The right panel shows that in the presence of Hiltonol (a dsRNA binding agent), PKR homodimerization occurs which leads to autophosphorylation of identified residues. This active form of PKR suppresses tumorigenesis. (**E**) A hypothetical model of the mechanisms by which Hiltonol mediates killing of lung cancer cells. Hiltonol effectively stimulates the expression of intracellular PKR and OAS [23], which are dsRNA-binding enzymes which mediate cellular antiviral and anti-tumor responses (Figure 4). PKR and OAS RNAi significantly restored cell survival and proliferation upon Hiltonol treatment. Consequentially, Hiltonol plays an immune-modulatory role, which suppresses pro-tumorigenic cytokines: GRO, IL-8, and IL-6, in H292 and H358, and MCP-1 in H1299. In addition, Hiltonol upregulates anti-tumor cytokine, IL-24, in A549 and H292, which also suggests that it contributes to the NSCLC tumorigenic microenvironment.

## Data Availability

Data available on request due to restrictions of privacy and ethical. The data presented in this study are available on request from the corresponding author. The data are not publicly available due to privacy and ethical issue.

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
