# Peer review of "Hiltonol Cocktail Kills Lung Cancer Cells by Activating Cancer-Suppressors, PKR/OAS, and Restraining the Tumor Microenvironment"

_ijms, 2021, doi:10.3390/ijms22041626_

Round 1

Reviewer 1 Report

Description of "Statistical analysis" section is very poor and need re-write.
For presented data Authors used two-way analyses and Student's t-test is not-correct tool for analysis.
Autors testing effects of dosage of Hiltonol from 10, 20, 50 to 100 on 94 four human NSCLC cell lines: A549, H292, H1299 and H358. Interaction between doses nad lines is very impotrent is these analyses but in reviewed paper lack of analysis of interaction.
In all figures need LSD values for comparison of presented results.
In my opinion paper needs major revision.

Reviewer 2 Report

The authors showed that Hiltonol (double-stranded polyriboinosinic-polyribocytidylic acid (poly IC) has anticancer activity in NSCLC. They related these activities to the modulation of the expression of different cytokines and the regulation of PKR and OAS, tumor suppression proteins which bind dsRNA. This work is potentially interesting, but several points need to be clarified and explored to better understand the article before to be published.

Major

  • In figure 2 and supplementary figure 1, the authors did not indicate the time point of apoptosis assay in the Annexin V/7AAD. How can the authors explain the ability of Hiltonol and Hiltonol+++ to induce caspase 3 activity, early apoptosis, and not late apoptosis?

To verify cell death induction, the authors may increase Hiltonol treatment time and perform AnnexinV/7AAD assay and PARP cleavage assay. Moreover, the authors should perform a senescence assay to exclude induction of senescence instead of cell death.

  • Figure 4 and related results are a little bit confusing. The authors showed a reduction in GRO, IL-8 and IL-6 in H538 cells, a reduction in MCP-1 expression in H1299 but an increase in the same cytokines in A549 and H292 cell lines. The authors hypothesized that the reduction of these cytokines could be related to the suppression of cell survival dependent by Hiltonol. It is not convincing. The authors should demonstrate this by knocking down the expression or treating the cell with these cytokines to counteract the increase and the decrease respectively, and measuring cell viability and cell proliferation.
  • Can the authors explain the reason why the different cell lines differently regulate cytokines expression after Hiltonol treatment?
  • Did the authors assess the effect of IL24 inhibition on the Hiltonol mediated effects on cell migration and invasion?  

Minor

  • In the 2.1 section, the authors defined as cell killing activity the reduction in cell viability measured by MTT or CTB assays. The definition is not correct because, in these experiments, the authors measured the metabolic activity of treated cells compared to control cells. The reduction in total metabolic activity in treated wells may be due to cell death and increased metabolically active cells in not treated controls. I suggest to replace the term cell death and cell killing with reduction in cell viability (lines 96, 102, 104, 106).
  • Lane 200 the authors should replace the sentence abrogate… with reduce Hiltonol-mediated reduction of cell viability.
  • FIGURE 5: did the author assess the protein expression level of PKR and OAS in control and siRNA treated cells?
  • Line 330: the authors should replace Immunofluorescent staining with IHC staining
  • Reference 3 and reference 10 are the same

Round 2

Reviewer 1 Report

Accept in present form.

Reviewer 2 Report

The authors have improved the manuscript according to the reviewers' suggestions. Although some issues need to be explored in future studies, the results presented are sound.

Regards,